# Performance of Reinforced Foam and Geopolymer Concretes against Prolonged Exposures to Chloride in a Normal Environment

**DOI:** 10.3390/ma16010149

**Published:** 2022-12-23

**Authors:** Muhammad Wasim, Rajeev Roychand, Rhys Thomas Barnes, Jason Talevski, David Law, Jie Li, Mohammad Saberian

**Affiliations:** 1Department of Civil & Infrastructure Engineering, RMIT University, Melbourne, VIC 3000, Australia; 2Department of Infrastructure Engineering, The University of Melbourne, Parkville, VIC 3010, Australia

**Keywords:** foam concrete, geopolymer concrete, reinforcement, chloride, corrosion, tensile strength, fractography, patch repairs

## Abstract

The utilization of sustainable cement replacement materials in concrete can control the emission of carbon dioxide and greenhouse gases in the construction industry, thus contributing significantly to the environment, society, and the global economy. Various types of sustainable concrete including geopolymer concrete are tested for their efficacy for construction in laboratories. However, the performance and longevity of sustainable concrete for civil engineering applications in corrosive environments are still debatable. This paper aims to investigate the performance of the reinforced geopolymer (GPC) and foam concretes (FC) against corrosive chloride exposure. Two long term key parameters, i.e., corrosion rate and mechanical performance of reinforcing steel in geopolymer and foam concrete were assessed to evaluate their performance against chloride attack. For experiments, reinforced GPC and FC specimens, each admixed with 3 and 5% chlorides, were kept at varying temperatures and humidity levels in the environmental chambers. The corrosion rates of the reinforced geopolymer and foam concrete specimens were also compared with control specimens after 803 days and the tensile strength of the corroded reinforcing steel was also determined. Moreover, the long term efficacy of repaired patches (810 days), in a chloride-rich surrounding environment utilizing FC and GPC, was investigated. The results suggested greater performance of FC compared to GPC under standard environmental conditions. However, the simulated patch repair with GPC showed better resistance against chloride attack compared to FC. The research also undertook the fractographical examination of the surfaces of the reinforcement exposed to 5% admixed chloride and develops models for the corrosion rates of foam concrete as a function of the corrosion rates of geopolymer concrete and chloride content. A correlation model for the corrosion rates of FC and GPC was also developed. The findings of the current research and the model developed are novel and contribute to the knowledge of long term degradation science of geopolymers and form concrete materials. Furthermore, the findings and methodology of the current research have practical significance in the construction and repair industry for determining the remaining service life for any reinforced and steel infrastructure.

## 1. Introduction

Large scale concreting produces emissions of greenhouse and carbon dioxide gases that impact the environment and subsequently cause socio-economic consequences [1,2]. The environmental regulatory bodies in every part of the world are emphasizing the use of environmentally friendly concrete for construction [3,4]. Therefore, researchers are investigating sustainable concretes such as geopolymer concrete and alkali-activated concretes [5] utilizing cement replacement and waste materials in construction [6,7,8,9,10,11,12,13,14,15,16,17]. However, the long-term durability of these sustainable concretes for construction in aggressive environments still need to be explored [18,19,20,21].

Foam concrete (FC), which can be classified as a geopolymer foam concrete, is a recently developed lightweight structural concrete that utilizes waste material in its composition [22]. The physical, mechanical, chemical, compositional, microstructural characteristics, and structural behavior of foam concrete have been investigated previously [23,24,25,26,27,28,29,30,31,32]. Additionally, the permeability and sorptivity of FC have been assessed by reviewers [33,34]. In addition, the durability of FC in the short-term (90 days) has been investigated recently [35]. However, at present, there are no research publications on the long-term corrosion behavior of FC indicating a gap for further new research.

On the other hand, fly ash- and slag-based geopolymer concretes (GPC) have been established for some time and researched extensively for their durability when incorporating various waste materials for structural and infrastructure applications [36,37,38,39,40]. Finite element simulations for the penetration of chloride ions into the geopolymer composite formwork [41] and the use of geopolymer concrete for resisting microbial corrosion has also been investigated [42]. Moreover, research on the use of GPC for structural repairs [43] and their passivating properties on the embedded reinforcement have been conducted [44]. However, the literature on the comparative durability of GPC [45], specifically foam-reinforced concrete is very limited. Furthermore, no long-term comparison of the mechanical behavior of the FC and GPC in chloride-rich environments has been carried out.

Therefore, there is from very limited to no research on the long-term corrosion resistance and the mechanical performance of the reinforced foam concrete in chloride-rich environments. In addition, a comparison of the corrosion resistance of foam concrete with geopolymer concrete has not been explored previously. The current research has been carried out to address this gap. Specimens of varying admixed chloride concentrations, along with control specimens of FC and GPC, were prepared and tested after 803 days of exposure for corrosion resistance and mechanical performance. In addition, the long-term efficacy of simulated patch repairs of FC and GPC with high chloride content in the surrounding concrete was investigated. Models are developed to predict the corrosion rates of FC as the function of the chloride content and the GPC corrosion rate and for the correlation between the two types of concretes.

## 2. Experimental Methodology

### 2.1. Specimens Preparation

The specimens of 180 × 75 × 75 mm^3^ were prepared for foam and geopolymer concrete with 3 and 5% chlorides, respectively. The control specimens of FC and GPC with no chlorides were also prepared for comparison and kept under standard conditions of 24 °C and 50% humidity. For the preparation of the FC specimens, an electronic mixer was used in which the required quantities (Table 1) of slag, fly ash, foam, and water with added chloride were thoroughly mixed. For the preparation of the foam, a uniform density of 70 kg/m^3^ was maintained. The detailed procedure for the preparation of the FC specimen was as previously reported [35]. The concrete was poured in the special molds having embedded reinforcing steel of 7 mm in diameter and 180 mm in length placed at the middle of the specimen. The typical composition of the steel by the weight percentage were C = 2.14, O = 0.67, Al = 0.05, Si = 0.08, Ti = 0.04, and Fe = 97.03. The sequential procedure adopted for the preparation of FC specimens is shown in Figure 1. A similar procedure of mixing was adopted for the GPC specimens with the additional coarse and fine aggregates along with 16M solutions of Na_2_SiO_3_ and NaOH. The mix design used for GPC is shown in Table 2. The test plan of the research is shown in Table 3. FC and GPC specimens after hardening were demolded and subsequently cured for 28 days in water. Then, they were transferred to the environmental chamber which were set at a temperature of 24 °C and 50% humidity. Since the objective of the current research was to determine the long-term performance of GPC and FC, the duration of 803 days was selected to observe substantial corrosion-induced degradation of reinforcing steel in these concretes. Besides, corrosion is generally a slow process, it requires time to initiate and accelerate. Therefore, based on experience and previous similar research, the exposure period of 803 days was thought to be appropriate for the current research.

Simulated patch-repaired specimens for each of the FC and GPC were also prepared. A special mold was developed for the preparation of the simulated patch-repaired specimens with three portions. The two end portions of the mold were filled with chloride (5%)-contaminated concrete (Figure 2a), while the middle section was filled with uncontaminated concrete after hardening of the end portions. After the setting of the center section (Figure 2b), all the specimens were kept in an environmental chamber for the duration of 810 days. For patch repair specimens both FC and GPC specimens were prepared in identical fashion.

The main objective of the current experimental program was to observe the degradation of the mechanical properties of the reinforcing steel in the GPC and FC specimen in the aggressive environments after long-term exposures. Based on the literature, it was hypothesized that the selected chloride levels, along with the temperature and humidity variation would substantiate the reduction in the tensile strength of the reinforcing steel in GPC and FC. Understanding the corrosion behavior and quantifying the reduction in tensile strength was planned to be the major outcome of the current research. Since, the tensile strength of the reinforcing steel is directly related to the service life of the reinforced concrete structure, therefore, observations of the tensile strength of reinforcing steel in GPC and FC in aggressive environments could assist for the accurate computation of the designed life of structures.

### 2.2. Tests for the Performance of FC and GPC

The performance of the reinforced FC and GPC in a corrosive environment is measured by conducting corrosion assessment, determining the ultimate strength of the reinforcement using tensile testing and fractographical examinations. The details of the tests are as follows:

#### 2.2.1. Corrosion Assessment

After 803 days, the specimens were broken open and the corroded reinforcement extracted for mass loss measurements. This involved fracturing the specimens and removing mortar from the bars. The weight of the reinforcing bar after removing the rust was recorded. This weight of the reinforcement is subtracted from the initial weight of the reinforcement prior to placement in the concrete. The difference of the two measurements is identified as the weight loss. The mass loss In grams was calculated for all the specimens after 803 days. ASTM G1-03 (2017) e1 [38] was adopted for the mass loss measurements. The rust cleaning solution was the mixture of 20 g antimony trioxide (Sb_2_O_3_) and 50 g stannous chloride (SnCl_2_) dissolved in 1 L hydrochloric acid (HCl, specific gravity 1.19) by stirring for a quarter of an hour at room temperature. The corroded reinforcing bars were diped in the prepared solution for approx. one minute and then removed. These bars were then dried and further cleaned using a brush. After cleaning, the bars were weighed. The mass losses were converted to the rates of corrosion [4,42,43,44] as follows:
(1)CR=(K×W)/(A×T×D)
where CR = corrosion rates in mm/year, K = 8.76 × 10^4^, W = mass loss in grams (initial weights—weights after corrosion), T = time in hours, A = total surface area in cm^2^, and D = density = 7.6 g/cm^3^

#### 2.2.2. Mechanical Performance

The mechanical performance of the reinforced concrete under corrosive conditions was evaluated by carrying out tensile testing. The tensile tests were conducted following a mass loss measurement with a universal testing machine. The rate of loading was maintained at 1.1 mm/min according to ASTM E8/E8M-16ae1.

#### 2.2.3. Fractography

The SEM images of the fractured areas of the corroded reinforcing steel embedded in a 5% chloride-rich environment after conducting tensile tests was performed. The purpose was to observe degradation in the microstructure of the steel due to corrosive environments after prolonged exposures.

## 3. Results

### 3.1. Time Dependent Degradation of Reinforced Foam Concrete

The reinforcement in the FC specimens were assessed for the extent of corrosion, Figure 3a. The bars exposed to chloride contamination were observed to be more corroded than those in the control FC specimens. The reinforcing bars after rust removal are shown in Figure 3b. The loss in weights of the reinforcing bars was converted to corrosion rates as per ASTM G1-03 (2017) e1 [46]. The corrosion rates of the reinforcing steel in FC with time are shown in Figure 4.

The corrosion rates in FC specimens of both lower (3%) and higher (5%) chloride content showed a decline with time (Figure 4). The corrosion rates reduced from 0.092 on the 30th day to 0.008 mm/year upon reaching 803 days of exposure for the 5% chloride admixed FC specimens. Similarly, for the 3% chloride admixed specimen, the corrosion rates were found to decrease from 0.053 to 0.004 mm/year. It is hypothesized that this could be due to cavitation in the microstructure of FC [35] or to the formation of a thick layer of rust on the reinforcing steel, which led to the reduction in the corrosion rates measurement from the 30th day. The formation of this layer of rust prevented the access of moisture and oxygen to the steel in the FC specimens.

#### Long-Term Corrosion Rates for FC

The corrosion rates of the FC specimens with zero chloride, 3%, and 5% Cl after 803 days of exposure are shown in Figure 5. The corrosion rates were found to increase three-fold and four-fold for the two chloride contents with respect to control specimen after 803 days.

### 3.2. Long Term Corrosion Rates of Steel in GPC

The corrosion rates of steel in GPC with varying quantities of chlorides was also determined. The visual observations of the corroded bars from GPC are shown in Figure 6. After mass loss measurements, the corrosion rates were found to be 0.25 and 3.16 times more for the 3 and 5% chloride-contaminated concrete with respect to control of the GPC indicating that over the longer term the corrosion rates increase in the GPC with time.

#### Comparison of FC and GPC

The comparison of long-term corrosion rates of FC and GPC specimens was also performed. This comparison is shown in Figure 4. The data demonstrate higher corrosion rates for FC when compared to GPC for 3% chloride but not for the higher, i.e., 5% chloride (Figure 7). This is again possibly due to the development of corrosion products in the higher chloride-contaminated FC that do not allow the diffusion of oxygen, moisture, and further chloride to the reinforcing steel resulting in the decrease in the corrosion rate observed.

Furthermore, the corrosion rates of steel in the control FC were found to be slightly lower than that in GPC. This can be attributed to the compact microstructure of FC as compared to the microstructure of GPC. GPC has larger pores and voids within the microstructure of the concrete through which oxygen and moisture can easily diffuse to the embedded reinforcing steel and initiate and propagate corrosion. Hence, the initial corrosion rates of reinforcing steel in GPC are slightly higher than FC (Figure 7).

### 3.3. Behavior of Simulated Corroded Repaired Patches with FC and GPC

The investigation of the repair efficacy using FC and GPC was also compared in the current research. The corrosion rates measured from the mass loss are shown in Figure 8 and 9. The corrosion rates of end portions were found to be 0.0085 and 0.0084 mm/year for reinforcing bars in FC. The corrosion rate of the middle section was found to be 0.003 mm/year, indicating the migration of chloride from chloride-contaminated end portions to the middle section (Figure 8).

However, the corrosion rates of the end sections of the GPC specimens were found to be 0.0073 and 0.008 mm/year, slightly lower than those of FC specimen (Figure 9). Additionally, the corrosion rate for the middle section of the GPC specimen was found to be 0.0016 mm/year, which was almost half that of the corrosion rate obtained for the same section in FC (see Figure 8). The result of the simulated repaired patch of the GPC indicates that very little chloride has diffused from the chloride-containing sections to the middle in the GPC as compared to the FC specimen. Thus, based on this result, it can be concluded that the use of the GPC for the patch repair can be more effective as compared to the FC.

### 3.4. Long Term Mechanical Performance of Corroded Steel in FC and GPC

#### 3.4.1. Reinforcement from FC Tested

The performance of a reinforced concrete structure can be determined based on the mechanical properties of the reinforcing steel [47]. Reinforcing steel in concrete structures has been found to degrade mechanically with time due to exposure to corrosive environments such as with chloride-induced corrosion of the reinforcing steel in concrete. In consequence, the life of the structure can be adversely affected [35,47]. Therefore, the tests for the tensile strength of the corroded reinforcing steel bars from FC and GPC specimens were undertaken.

The reinforcement bars removed from the control and chloride-contaminated FC specimens showed some degradation of mechanical properties especially in the ultimate tensile strength for the highest chloride content specimens after 803 days (Figure 10). The ultimate tensile strength was found to be 536.2 MPa for the reinforcing bar in the 5% chloride-contaminated specimen (Figure 10a) that was approximately 3% lower than the ultimate strength of the reinforcing steel in the control specimens. The ultimate tensile strength was found to be 552.2 Mpa for the reinforcing steel in the control FC specimen (Figure 10c). The reduction in the ultimate strength of the reinforcing steel of 5% chloride-contaminated specimens could be due to the development of small pits and intergranular cracking, which led to the failure of the reinforcing steel at lower loads as compared to the reinforcing steel in the controlled specimen with no pits developed after 803 days.

There was no change found in the ultimate strength of the reinforcing steel in 3% chloride FC (Figure 10b). There was no variation in the failure strains (ductility) of the reinforcing steel under ambient conditions, even in the presence of the high chloride content of 5% after 803 days of exposure.

Thus, based on the stress and strain curves of reinforcing steel obtained for controlled, 3%- and 5%-contaminated chloride FC, it is inferred that the there is no significant corrosion-induced degradation of the reinforcing steel in high chloride-contaminated FC specimens after 803 days. More prolonged exposures and the higher temperatures such as 50 °C might have induced significant degradation in the mechanical properties of the reinforcing steel of FC. This can be explored along with the investigation of the porous structures of the FC as a scope of future research that can reveal the long-term performance of these concretes.

#### 3.4.2. Reinforcement from GPC Tested

The reinforcement from the GPC specimens was also tested for tensile strength. The ultimate tensile strength of the reinforcement for the 5% chloride specimen was found to be 529.6 MPa (Figure 11). The ultimate tensile strengths for the reinforcement in the 3% and control specimens were found to be 544 Mpa and 556.21 Mpa, respectively. The failure strains for the reinforcement in 5%, 3%, and control GPC specimens were found to be 0.075, 0.079, and 0.077. As observed in the FC specimens, very little or marginal variation in the failure strains of the reinforcement in GPC was observed owing to the same reason as explained for the high chloride-contaminated FC specimens. Further prolonged experimentations and pore structure investigations of the GPC are needed to observe significant degradation of the mechanical properties of reinforcing steel in GPC.

#### 3.4.3. Comparison of Mechanical Performance of FC and GPC

Upon comparison of the tensile strength results of the reinforcing bars from FC and GPC specimens, no major variations are observed. The reduction in the ultimate strength of the steel was slightly greater in the 5% FC (Figure 10a) as compared to the same GPC (Figure 11a). The ultimate strength of the reinforcement for the remaining specimens in FC and GPC were comparable. In addition, the trends of the failure strains for the reinforcements in both the FC and GPC were similar.

## 4. Discussion

The long-term durability and mechanical performance of FC against the chloride attack has not been established. The current research has shown that the corrosion rates of reinforcing steel in FC can be higher initially (Figure 4). However, the reinforcing steel remains protected over the longer period with the corrosion rate reducing with time. There could be several possible reasons for the reduction in corrosion rates. It could be the development of a thick layer of corrosion products on the reinforcing steel, due to the initial high corrosion rate, preventing access to oxygen and water or the reformation of the protective passive layer on the reinforcing steel. Given that the long-term rates for both the 3% and 5% chloride are significantly lower as compared to their respective initial 30th day measurements, the utilization of FC is advised for applications where the possibility of high chlorides exposure on the infrastructure is very high.

The reinforcement for the simulated repair (no chloride) of the FC specimen showed higher corrosion rates (Figure 8) than the control FC specimen (Figure 5). This is attributed to the migration of chloride from the chloride-contaminated concrete to the central chloride free patch. This migration resulted in the chloride-induced corrosion of the reinforcing steel, thus the corrosion rates of the simulated repair section was found slightly higher than the control specimens after 810 days of exposure.

The corrosion rates of the reinforcing steel in the control and chloride-contaminated GPC specimens were found to be slightly higher than in FC. The formation of dense and compacted microstructures and the resulting refined pore structure is hypothesized as a barrier for the diffusion of oxygen and moisture to the reinforcement, subsequently, the corrosion rates decreased, even in the high chloride surrounding environment [48]. Alternatively, it could be due to the redevelopment of a passive layer on the reinforcing steel that prevented the rise in corrosion rates due to on-going repolymerization re-passivating the steel [49]. These results suggest that over a longer time frame FC is more resilient than the GPC specifically for construction in high chloride and marine environments.

The patch repairs with GPC and FC concrete showed the efficacy of GPC over FC as the corrosion rates of the simulated patch repair of GPC were found to be significantly lower than those of FC (Figure 8). These findings suggest the use of GPC, or even FC, could be extremely useful for the repair in structural and infrastructure applications. Patch repairs with ordinary reinforced concretes in the surrounding of similar chlorides contaminations have been found to develop a macro-cell with high corrosion rates in repaired patches due to the separation of anodes and cathodes [50]. However, with FC and GPC, the phenomenon of macro-cell formation was not visible over the duration of the testing.

The mechanical behavior of the reinforcing steel in FC and GPC with high admixed chloride content showed similar levels of degradation with time in ultimate tensile strength tests. The reinforcement in GPC was slightly more degraded than that in FC. The corrosion rates of the GPC bars were also found to be slightly higher than those of FC. This would indicate that the mechanical performance of reinforcing bars in FC is superior to those in GPC when exposed to high levels of chloride.

### 4.1. Fractographic Examination of the Corroded Reinforcement

The fractographical examination of the corroded bars was also carried out to determine the cause of the degradation in the ultimate strength of the reinforcement in 5% chloride FC and GPC specimens. The SEM images of the fractured surface of the reinforcement were taken from low to high magnifications with a spot size of 5 and WD of 10, as shown in Figure 12. Figure 12a represents the overall fractured section after tensile testing of the reinforcing bar in 5% chloride specimens. Figure 12b represents the SEM image of the fractured surface at a magnification of 500× highlighting a ferritic microstructure (composition show 97% Fe). Steel reinforcement is predominantly ferrous metals with a ferritic microstructure, which is susceptible to corrosive elements [51]. Figure 12c, with a magnification of 1000×, indicates the presence of cracks within the microstructure, which, at a higher magnification of 2000× (Figure 12d), reveal the presence of multiple microcracks and clusters in the form of branching including micro-voids.

Intergranular cracks and micro-voids have been found to degrade the mechanical properties of steel in highly aggressive environments of chloride and hydrogen [47,52]. The coupled effect of chloride corrosion and hydrogen gas released from the corrosion reaction diffusing into the microstructure of the reinforcing steel could have both contributed to the degradation in the ultimate strength observed.

The fractography investigation confirms that it was the intergranular cracking and the formation of micro voids that led to the degradation of the ultimate strength of the reinforcing steel in the high chloride-contaminated FC and GPC specimens.

### 4.2. Models for the Long-Term Corrosion Rates of FC

A model for the corrosion rates of reinforcement FC as the function of chlorides content and the corrosion rate of the GPC determined was developed based on the test data, with a high R^2^ and a low standard error estimate. Based on the data, the following model was developed:
(2)FCcr =0.004 × Cl −3.101 × GPCcr +0.01
where FCcr = corrosion rates of foam concrete, Cl = Chloride, and GPCcr = Geopolymer concrete corrosion rates.

The Pearson correlation between the variables was also determined, the results are shown in Table 4. The results indicate strong correlation between FCcr and chloride, as well as with GPCcr. Both the concretes showed an increase in corrosion rates with the high chloride content.

A correlation model between the corrosion rates of the FCcr and GPCcr was also developed with R^2^ = 0.84 and very low standard error estimate. The developed model is as follows:(3)FCcr =0.893 × GPCcr + C 
where C = constant = 0.00029.

The developed models will have practical significance in the field for specifying concrete. Furthermore. with the addition of more variables such as time, temperatures, and humidity, the models with applications in cold and hot environments can be developed.

## 5. Conclusions

The current paper has presented a comparative analysis of the performance of reinforced foam and geopolymer foam concrete against severe chloride attack under standard environmental conditions. Specimens with 3% and 5% chloride containing reinforcement bars were tested for the corrosion rate and tensile strength at 803 days. Their corrosion rates and mechanical behavior were compared with respective control specimens without chlorides. Simulated repaired patches of FC and GPC were also investigated for their efficacy in high chloride applications. Based on the results, the following findings and recommendations can be summarized:The corrosion rates were found to increase three-fold and four-fold for 3 and 5% admixed chloride content FC specimens as compared to the control specimen with 0% chloride after 803 days, respectively.The specimens with higher (5%) chloride contents suffered degradation of the tensile strength of their reinforcements. The tensile strengths of the reinforcements embedded in 5% chloride-contaminated FC and GPC specimen were found to be 536.2 and 529.6 MPa, respectively. The tensile strengths of reinforcement in their control specimens were found to be 552.2 and 556.21 MPa, respectively.FC showed more resilience as compared to the GPC, with respect to both the corrosion rate and mechanical behavior after 803 days. The corrosion rates of FC were found to be 5% lower than those of GPC. The degradation of reinforcement was found to be 2.89% in FC, while in GPC it was found to be 4.78% as compared to the control specimen.The fractography of the reinforcing steel indicated the presence of microcracks that induced degradation in the ultimate tensile strength. It was probably due to the chloride-enhanced hydrogen embrittlement that caused the formation of microcracks within the microstructure of the reinforcing steel in FC and GPC.A model for the long-term corrosion rates of FC as the function of the corrosion rates of GPC and chloride content was developed with a very high R^2^ and low standard error estimate. A correlation model for the corrosion rates between FC and GPC was also developed with R^2^ = 0.84.The simulated repaired patches of reinforced FC showed slightly higher corrosion rates as compared to the GPC. The corrosion rates of the repaired section with FC were twice than those of GPC after 810 days. This was probably due to the migration of chloride, oxygen, and moisture from the surrounding chloride-rich environment to the reinforcing steel in the repaired section in FC due to its microstructure and cavities as compared to GPC.The test results suggest that the test duration could be even longer for such experiments to observe significant variation of the mechanical properties. Moreover, investigations for the porous structure and solid phases of the GPC and FC should be carried out as the scope of future research.

Finally, to conclude, the results presented in the research have practical significance. With further testing and the incorporation of more variables, very accurate models for the degradation of FC and GPC can be developed.

## Figures and Tables

**Figure 1 materials-16-00149-f001:**
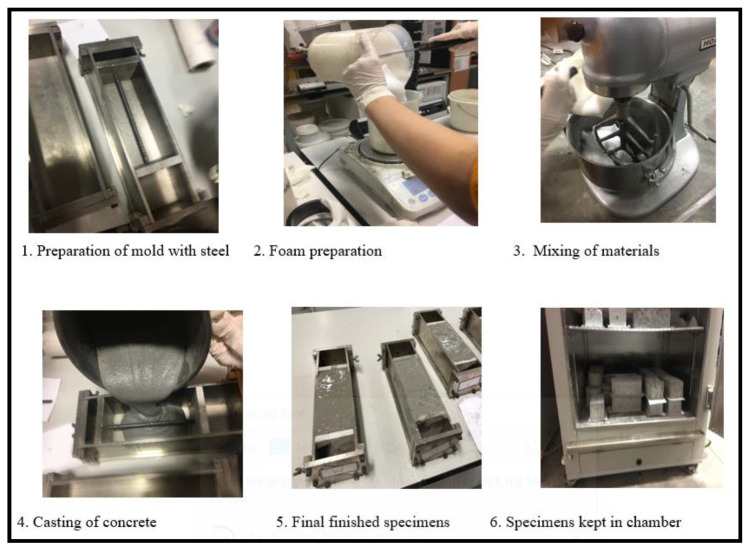
Illustration for the preparation of specimens.

**Figure 2 materials-16-00149-f002:**
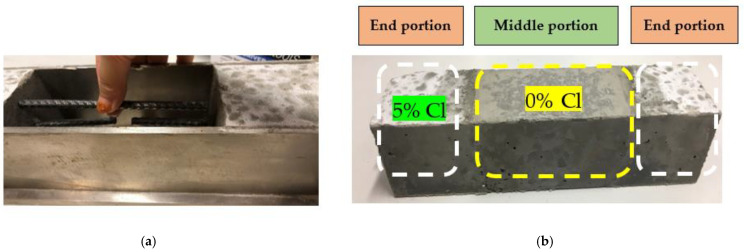
Simulated repaired specimen (**a**) placement of reinforcement and then filling of middle section of the specimen with controlled concrete with no chloride; (**b**) freshly hardened middle section simulating repairs.

**Figure 3 materials-16-00149-f003:**
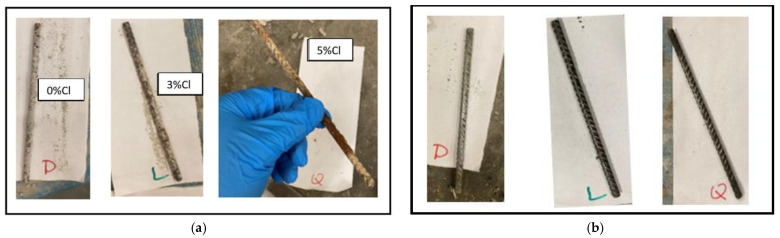
Reinforcing bars extracted from the controlled D—0 chloride, L—3% chloride, and Q—5% chloride admixed FC specimens: (**a**) Corroded; (**b**) After rust cleaning.

**Figure 4 materials-16-00149-f004:**
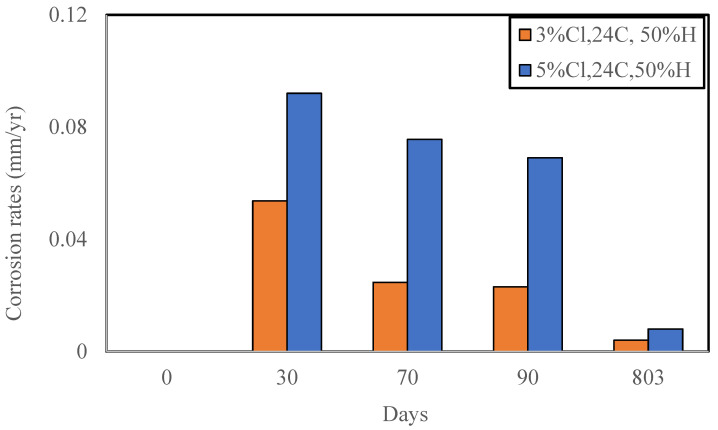
Time-dependent corrosion rates of FC specimens.

**Figure 5 materials-16-00149-f005:**
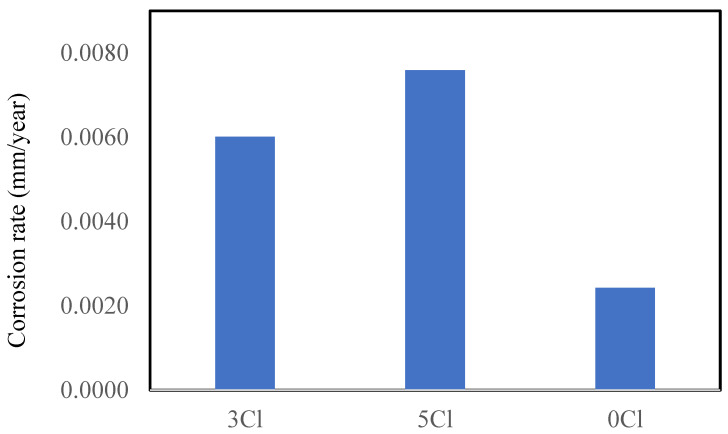
Corrosion rates of the FC specimens after 803 days in varying admixed chloride quantities.

**Figure 6 materials-16-00149-f006:**
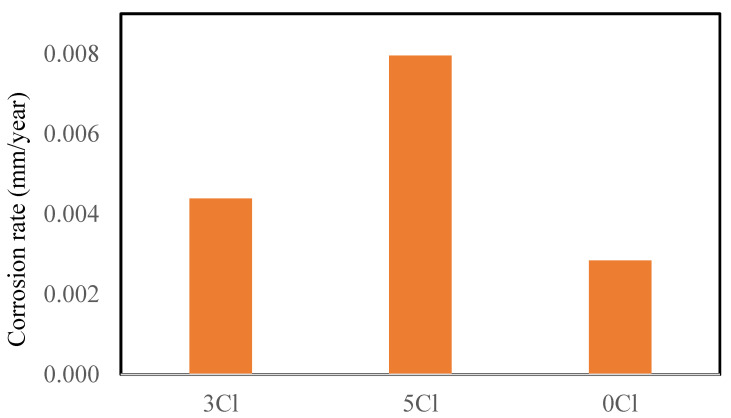
Corrosion rates of the GPC specimens after 803 days in varying admixed chloride quantities.

**Figure 7 materials-16-00149-f007:**
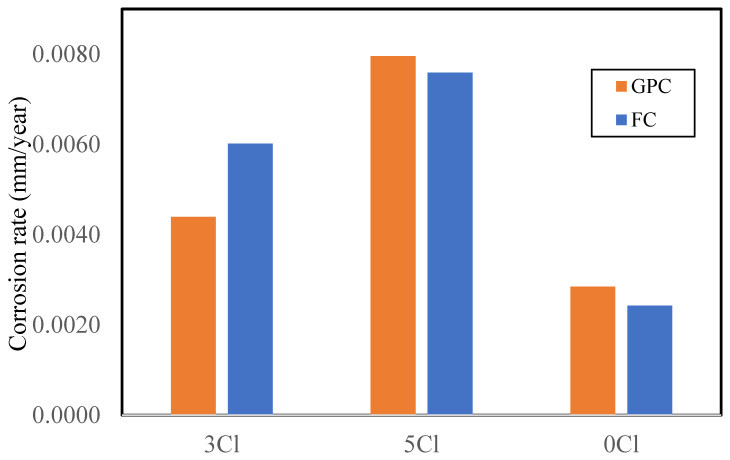
Comparison of corrosion rates of GPC and FC after 803 days admixed with varying chloride quantities.

**Figure 8 materials-16-00149-f008:**
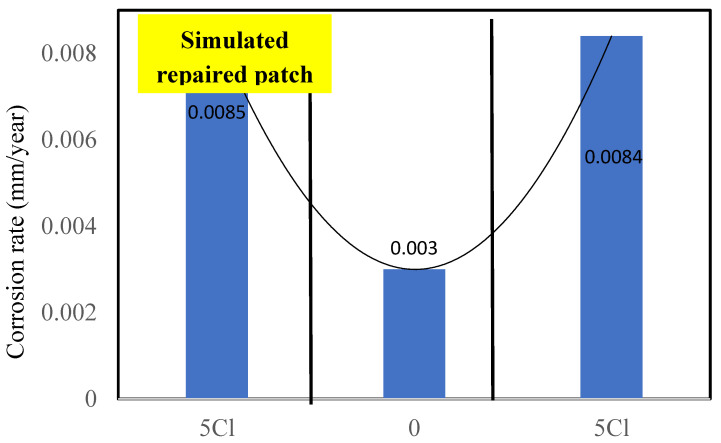
Corrosion rate of simulated refurbished repaired patches of the GPC concrete where zero in the middle indicated no chloride after 810 days.

**Figure 9 materials-16-00149-f009:**
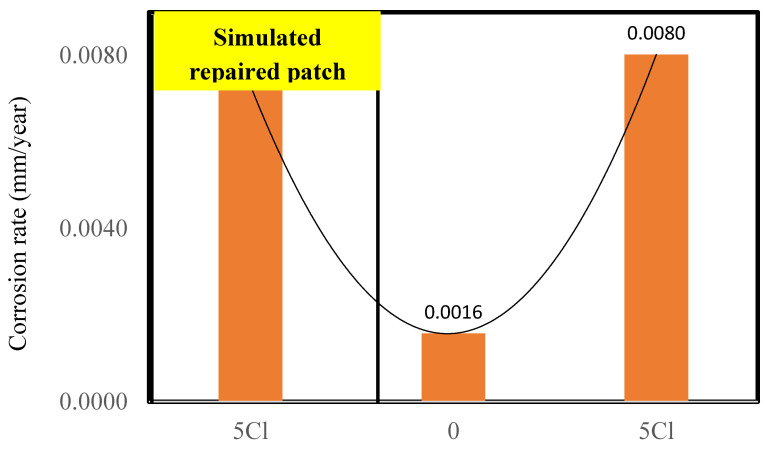
Corrosion rates of simulated refurbished repaired patches of the GPC concrete where zero in the middle indicated no chloride after 810 days.

**Figure 10 materials-16-00149-f010:**
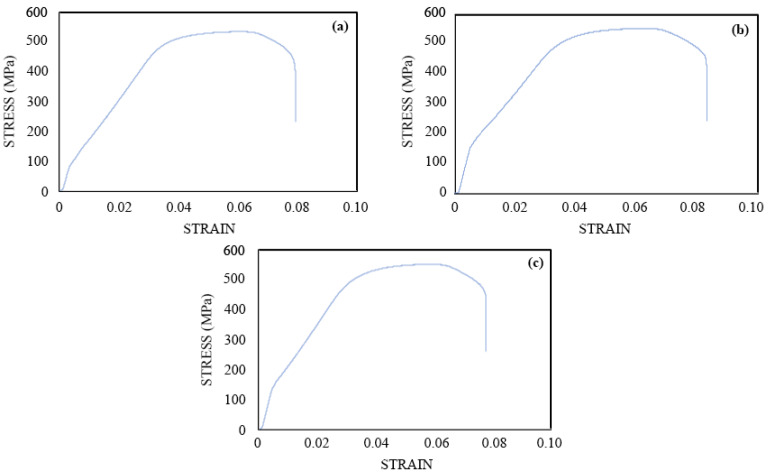
Stress–strain curves for the reinforcing steel in various FC specimens, (**a**) 5%; (**b**) 3%; (**c**) 0% after 803 days of exposure.

**Figure 11 materials-16-00149-f011:**
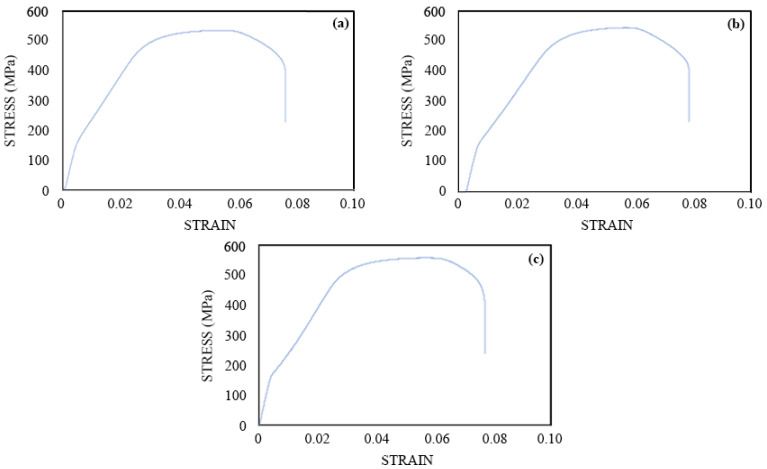
Stress–strain curves for the reinforcing steel in various GPC specimens, (**a**) 5%; (**b**) 3%; (**c**) 0% after 803 days of exposure after 803 days of exposure.

**Figure 12 materials-16-00149-f012:**
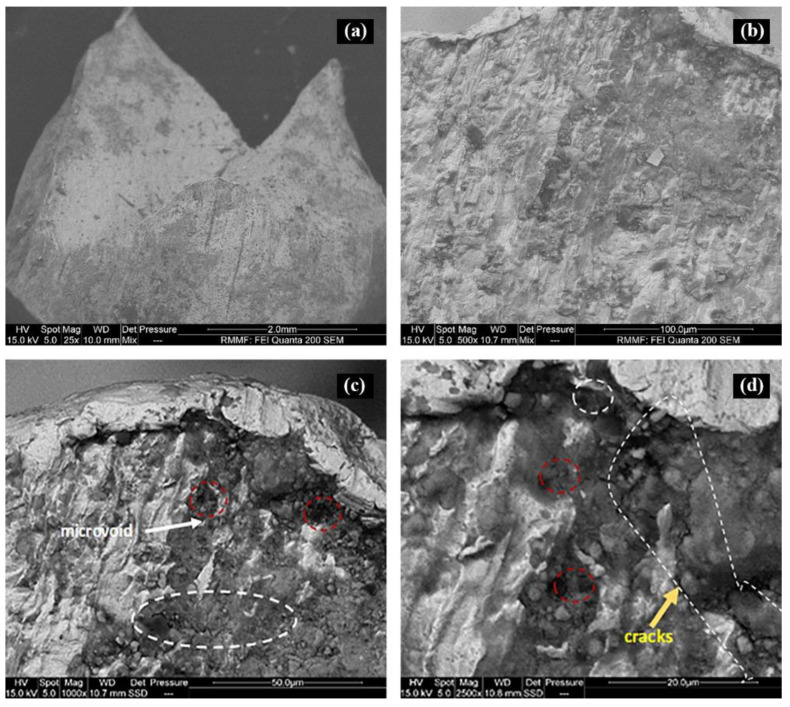
The microstructure of the fractured surface of the corroded bar (**a**) from low (**b**) to high magnifications (**c**,**d**) with cracks indicated by white dotted lines and the micro voids by red dotted lines.

**Table 1 materials-16-00149-t001:** Mixture composition for FC specimens.

Foam of Sodium Silicate (g)	Fly Ash (g)	Slag (g)	Water (g)	NaCl (g)	
				3% Cl	5% Cl
50	200	200	160	20	33.3

**Table 2 materials-16-00149-t002:** Mixture composition for GPC specimens.

Coarse Aggregate of 14 mm (kg/m^3^)	Sand (kg/m^3^)	Fly Ash (kg/m^3^)	Slag (kg/m^3^)	Water (kg/m^3^)	NaCl (kg/m^3^)	
					3% Cl	5% Cl
1071	526	196	196	196	17.25	29

**Table 3 materials-16-00149-t003:** Test plan.

Specimen	Type of Concrete	Temperature (°C)	Humidity (%)	Duration (days)	Testing/ Measurement
3% Cl admixed (L)	FC	24	50	803	Corrosion rates and Tensile strength
	GPC	24	50	803	
5% Cl admixed (Q)	FC	24	50	803	
	GPC	24	50	803	
Controlled (D)	FC	24	50	803	
	GPC	24	50	803	
Simulated patch repair	GPC—Left End Section (5% Cl)	24	50	810	
	GPC—Middle Section (0% Cl)	24	50	810	
	GPC—Right End Section (5% Cl)	24	50	810	
	FC—Left End Section (5%Cl)	24	50	810	
	FC—Middle Section (0% Cl)	24	50	810	
	FC—Right End Section (5% Cl)	24	50	810	

**Table 4 materials-16-00149-t004:** Pearson correlation between dependent and independent variables.

		FC	GPC	Cl
Pearson Correlation	FCcr	1.000	0.902	0.943
	GPCcr	0.902	1.000	0.994
	Cl	0.943	0.994	1.000
Sig. (1-tailed)	FCcr	0.0	0.142	0.108
	GPCcr	0.142	0.0	0.034
	Cl	0.108	0.034	0.0
N	FCcr	3	3	3
	GPCcr	3	3	3
	Cl	3	3	3

## Data Availability

All data, and models, generated or used during the study appear in the submitted article.

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
