# Peer review of "Performance of Reinforced Foam and Geopolymer Concretes against Prolonged Exposures to Chloride in a Normal Environment"

_materials, 2022, doi:10.3390/ma16010149_

Round 1

Reviewer 1 Report

Dear Authors, there is no intro in your paper as such. You need to elaborate one with concrete data and reflection of work done, underline the novelty of your study. Particular reason why the rest duration is 803 days or 810? And the intermediate results, any? Please elaborate also structure of your paper, sometimes it has no clearance. Indicate the goal and objectives. 

Author Response

Comment 1: Dear Authors, there is no intro in your paper as such. You need to elaborate one with concrete data and reflection of work done, underline the novelty of your study. Particular reason why the rest duration is 803 days or 810? And the intermediate results, any? Please elaborate also structure of your paper, sometimes it has no clearance. Indicate the goal and objectives.

Response 1:

Thank you for your comment. We have incorporated following changes into the revision.

  • We have added the significance and novelty of the current research in the Abstract.
  • Added new and relevant references as suggested by other reviewers in the Introduction section.
  • Indicated the goals and objectives of the current research in the Methodology.
  • Highlighted the reason for the selection of duration for the experimentation in the Methodology.
  • Further elaborate results in the lights of other reviewers’ comments.
  • Improved the Conclusions section.
  • Revised the Reference style and updated them.

Reviewer 2 Report

Please further describe the main steps that you followed and the main outstanding outcomes in the abstract.

Please further elaborate on the novelty of your work in the abstract.

The present form of introduction is pretty modest. Please revise the latest related studies to your work. Accordingly, please include the research works in the last five years. 

Please include a short summary of the work titled Predicting the Compressive Strength of Concrete Containing Binary Supplementary Cementitious Material Using Machine Learning Approach. Accordingly, please explain the effect of adding different additives on improving the mechanical properties of concrete.

Please delineate the material properties and concrete properties used in this research.

Please further explain the failure criteria for the performed experiments.

Please include the statistical characteristics of the reported results in Figures 4 and 5.

Please add a compelling discussion on the effect of key parameters that can affect the outcome of your study.

Please clarify the captured images in Figure 12.

Please add a compelling discussion on the most important parameters that can significantly affect the reported outcomes in this work.

Please revise the conclusion and present a more condensed version including the major results of this research, limitations, and recommendations for future work.

Author Response

Comment 1: Please further describe the main steps that you followed and the main outstanding outcomes in the abstract.

Response 1: Thank you for the comment.

We have updated the main steps and the outstanding outcomes in the Abstract. Please check the highlighted text (blue) in the revised manuscript.

Comment 2: Please further elaborate on the novelty of your work in the abstract.

Response 2: Thank you for the comment.

We have updated the novelty of the research in the Abstract. Please check the highlighted text in the revised manuscript.

Comment 3: The present form of introduction is pretty modest. Please revise the latest related studies to your work. Accordingly, please include the research works in the last five years.

Response 3: Thank you for the comment.

We have updated the Introduction section with new references. Please check the revised manuscript.

Comment 4: Please include a short summary of the work titled Predicting the Compressive Strength of Concrete Containing Binary Supplementary Cementitious Material Using Machine Learning Approach. Accordingly, please explain the effect of adding different additives on improving the mechanical properties of concrete.

Response 4:  Thank you for your comment. We have referred the mentioned publication in the introduction section of the manuscript. However, the intension of our work was to find the effect of aggressive environment on the corrosion and mechanical properties of the reinforcing steel in geopolymer and foam concrete. Therefore we have not conducted in compressive strength tests in the current research. However, as suggested by reviewer, in future we have planned to conduct more experiments in which we take into account the compressive strength of the reinforced concrete. Hope it answers the worthy reviewer’s query.

Comment 5: Please delineate the material properties and concrete properties used in this research.

Response 5: Thank you for the comment.

Through this research, we wanted to explore the influence of chlorides, temperature and humidity on the corrosion and the mechanical properties of the reinforcing steel. Therefore, we did not conduct the testing for the materials and concrete properties in this research. However, for future research and extension, we have planned to use new materials and concrete testing.

Many thanks 

Comment 6: Please further explain the failure criteria for the performed experiments.

Response 6:  Thank you for the comment.

We have further improved the failure criteria for the experiments in the revision. We have added the following lines in the revised manuscript for failure criteria:

“The main objective of the current experimental program was to observe the degradation of the mechanical properties of the reinforcing steel in the GPC and FC specimen in the aggressive environments after long-term exposures. Based on literature, it was hypothesized that the selected chloride levels, along with temperature and humidity variation would substantiate the reduction in the tensile strength of the reinforcing steel in GPC and FC. Understanding the corrosion behaviour and quantifying the reduction in tensile strength was planned to be the major outcome of the current research. Since, tensile strength of the reinforcing steel is directly related to the service life of reinforced concrete structure, therefore, observations of the tensile strength of reinforcing steel in GPC and FC in aggressive environments could assist for accurate computation of designed life of structures.”

Comment 7: Please include the statistical characteristics of the reported results in Figures 4 and 5.

Response 7: Thank you for the comment.

We have now included the data labels in the revised figures 4 anf 5.  Please check the revised manuscript.

Comment 8: Please add a compelling discussion on the effect of key parameters that can affect the outcome of your study.

Response 8: Thank you for the comment.

The detailed discussion is added in the revision in Section 3.4 and also in Section 5.1 of the revised manucript.

Comment 9: Please clarify the captured images in Figure 12.

Response 9: We have clarified the captured images in Figure 12 in the revised manuscript. The caption is revised and the conclusive statement is added in the last paragraph of Section 5.1of the revised manuscript to further clarify images in Figure 12.

Comment 10: Please add a compelling discussion on the most important parameters that can significantly affect the reported outcomes in this work.

Response 10:  The detailed discussion on the parameters is now added. Thank you

Comment 11: Please revise the conclusion and present a more condensed version, including the major results of this research, limitations, and recommendations for future work.

Response 11: We have revised the conclusion section to suit the reviewer’s comment.

Reviewer 3 Report

The paper only presents the chloride test, mechanical test and SEM. Authors claimed that “This was probably due to the migration of chloride, oxygen and moisture from the surrounding chloride rich environment to the reinforcing steel in repaired section in FC due to its microstructure and cavities as compared to GPC”. Therefore, some pore structure test should be carried out.

The differences between the different mixtures should be verified from the pore and solid phases.

Besides, some figures should be combined together, such as the Figure 11. Different strain curves in Figure 11 are very similar, are there only three curves? What about the data fluctuation? More explanations should be added.

The self-citation is serious. Authors should concern relevant studies from other research groups, such as Effects of fineness and content of phosphorus slag on cement hydration, permeability, pore structure and fractal dimension of concrete; The influence of fiber type and length on the cracking resistance, durability and pore structure of face slab concrete. Besides, The format of references is messy. It should be polished.

Author Response

Comment 1: The paper only presents the chloride test, mechanical test and SEM. Authors claimed that “This was probably due to the migration of chloride, oxygen and moisture from the surrounding chloride rich environment to the reinforcing steel in repaired section in FC due to its microstructure and cavities as compared to GPC”. Therefore, some pore structure test should be carried out.

Response 1: Thank you for the comment. Authors agreed to the worthy reviewer about the pore structure tests. This is an ongoing work and we will incorporate these tests in the extension of the current research.

Comment 2: The differences between the different mixtures should be verified from the pore and solid phases.

Response 2:  Thank you the good comment. This will be the scope of our future works.

The current work was to find the performance behaviour of GFC and FC under aggressive environments for long term which has no previous research. We have highlighted the need of more prolonged testing and pore and solid phases investigations as a scope of future research in the last point of the Conlusions

Comment 3: Besides, some figures should be combined together, such as the Figure 11. Different strain curves in Figure 11 are very similar, are there only three curves? What about the data fluctuation? More explanations should be added.

Response 3:  Thank you for the comment.

We have added the detailed discussions including data fluctuation and relevant explanations. Please consider the highlighted lines in Section 3.4 of the revised manuscript.

Comment 4: The self-citation is serious. Authors should concern with relevant studies from other research groups, such as Effects of fineness and content of phosphorus slag on cement hydration, permeability, pore structure and fractal dimension of concrete; The influence of fiber type and length on the cracking resistance, durability and pore structure of face slab concrete. Besides, The format of references is messy. It should be polished.

Response 4: Thanks Thank you for the comment. We have incorporated the references in the Introduction section of the revised manuscript to address the reviewer’s comment.

Round 2

Reviewer 1 Report

Please highlight all the changes. Consider to enlarge and to elaborate in detail the points mentioned before. Why test lasts 803 days? 

Author Response

Please note that all the reviewer's comments have been addressed in the last submission with the track changes and highlights. Our last submission response (as below):

We have incorporated following changes into the revision.

  • We have added the significance and novelty of the current research in the Abstract with track changes
  • Added new and relevant references as suggested by other reviewers in the Introduction section.
  • Indicated the goals and objectives of the current research in the Methodology.
  • Highlighted the reason for the selection of duration for the experimentation in the Methodology.
  • Further elaborate results in the lights of other reviewers’ comments.
  • Improved the Conclusions section.
  • Revised the Reference style and updated them.

Reviewer 2 Report

N/A

Author Response

We thank the reviewer for the positive feedback

Reviewer 3 Report

It can be accepted now.

Author Response

Comment: N/A

Response: We thank the reviewer for accepting our response